# Whole Exome Sequencing Study Suggests an Impact of *FANCA*, *CDH1* and *VEGFA* Genes on Diffuse Gastric Cancer Development

**DOI:** 10.3390/genes14020280

**Published:** 2023-01-21

**Authors:** Alfiia Nurgalieva, Lilia Galliamova, Natalia Ekomasova, Maria Yankina, Dina Sakaeva, Ruslan Valiev, Darya Prokofyeva, Murat Dzhaubermezov, Yuliya Fedorova, Shamil Khusnutdinov, Elza Khusnutdinova

**Affiliations:** 1Federal State Budgetary Educational Institution of Higher Education, Ufa University of Science and Technology, 450076 Ufa, Russia; 2Institute of Biochemistry and Genetics, Ufa Federal Research Center of the Russian Academy of Sciences, 450054 Ufa, Russia; 3Federal State Educational Institution of Higher Education, Bashkir State Medical University, 450008 Ufa, Russia

**Keywords:** gastric cancer, whole exome sequencing, germline mutations, somatic mutations, pathogenic variants

## Abstract

Gastric cancer (GC) is one of the most common cancer types in the world with a high mortality rate. Hereditary predisposition for GC is not fully elucidated so far. The aim of this study was identification of possible new candidate genes, associated with the increased risk of gastric cancer development. Whole exome sequencing (WES) was performed on 18 DNA samples from adenocarcinoma specimens and non-tumor-bearing healthy stomach tissue from the same patient. Three pathogenic variants were identified: c.1320+1G>A in the *CDH1* gene and c.27_28insCCCAGCCCCAGCTACCA (p.Ala9fs) of the *VEGFA* gene were found only in the tumor tissue, whereas c.G1874C (p.Cys625Ser) in the *FANCA* gene was found in both the tumor and normal tissue. These changes were found only in patients with diffuse gastric cancer and were absent in the DNA of healthy donors.

## 1. Introduction

Gastric cancer (GC) is one of the leading causes of death in the world. In the Russian Federation, cancer of this localization ranks sixth among all malignant tumors in terms of incidence and second in terms of mortality [1]. Known inherited predisposition to malignant neoplasms of the stomach with a high risk of 70–83% accounts for mutations in a number of genes, including *CDH1, TP53, MLH1, MSH2*, and others [2]. However, only a small proportion of gastric adenocarcinomas occur within the framework of a clear hereditary component. Only about 5–10% of patients have a burdened family history; rather, in most cases sporadic gastric cancer occurs. Genomic studies can reveal other new genes and genetic variants underlying disease [3]. Whole exome sequencing makes it possible to identify new genetic risk variants as germline mutations or polymorphic sites, which could be associated with gastric cancer, as well as somatic mutations or microsatellites and copy number variations, which can be of key importance for early diagnosis, prognosis and treatment of the disease [4].

In 2011, Wang K. and colleagues were among the first to publish the results of whole exome sequencing using the Next-Generation Sequencing (NGS) technology of twenty-two DNA samples from GC patients and found a high mutation rate of genes encoding proteins involved in chromatin remodeling processes [5]. Further studies on genetic determinants of GC development have indicated numerous genes, such as *ARID1A, MLL3, FAT4, PIK3CA* and *MLL*, to be involved in the etiology of the disease. The *FAT4* and *ARID1A* genes, for instance, might be candidate tumor suppressors, and if inactivated may promote the pathogenesis of various GC subtypes [6,7]. Lee H.H. et al. studied the mutational spectra in primary gastric cancer and the corresponding metastases to the lymph nodes and found molecular changes in the *SMARCA4* gene as a late event in primary tumors. The authors also discovered that mutations in the *CTNNB1* gene are specific for malignant stomach tumors that metastasize to the lymph nodes [8]. NGS based analysis revealed the high number of somatic mutations, including changes in the previously described genes *TP53, ARID1A, FAT4, LRP1B, PTPRT, FAT1, APC* and others, per gastric cancer patient [9]. Advances in NGS technologies have resulted in uncovering of genotype-phenotype associations for many tumor types, including GC, but only some gastric tumors could be explained by mutations in known or recently described genes. Identification of the variants causing the disease brings the research into clinical practice.

To identify novel genetic risk variants underlying GC etiopathology, we applied WES analysis in a group of Russian patients with GC, followed by replication of our results for three identified pathogenic variants in a cohort of 30 patients with diffuse gastric cancer and 30 healthy donors. This study reports our experience using WES to discover novel, coding variants, likely responsible for the progression of GC.

## 2. Materials and Methods

### 2.1. Patient Samples

We used DNA samples isolated from peripheral blood (*n* = 200), as well as tumor (*n* = 70) and adjacent normal gastric tissue (*n* = 9), obtained during the surgery in patients with a histologically confirmed diagnosis of GC, who are being treated at the Republican Clinical Oncological Dispensary, Ufa, Russia during the period years 2017–2020. The control group included 200 unrelated subjects showing no signs of gastropathology. Diagnostic criteria included anamnesis data, physical examination laboratory and instrumental examinations, as well as pathological and anatomical examination data, adopted in accordance with clinical guidelines developed jointly by the All-Russian National Union “Association of Oncologists of Russia” and the All-Russian public organization “Russian Society of Clinical Oncology”. The sampling was carried out by the staff of the Surgical Department No. 1 of the Republican Clinical Oncological Dispensary in Ufa in accordance with the ethical standards of the bioethical committee, based on the Helsinki Declaration of the World Medical Association, “Ethical principles for conducting scientific medical research involving a person as a subject.” Samples for whole exome sequencing (*n* = 18 in total) included DNA samples isolated from tumor (*n* = 9) and non-tumor-bearing healthy gastric tissue of patients with gastric adenocarcinoma stage III, group 2 (*n* = 9). All tumors were classified according to the TNM system consistent with the requirements of the International Cancer Union. The age of patients ranged from 45 to 77 years old, and the average age of disease manifestation was 63.44 years. A more detailed description of the samples is presented in Table 1 (Table 1). Genomic DNA was isolated by routine phenol-chloroform extraction.

### 2.2. Methods

#### 2.2.1. Whole Exome Sequencing and Data Analysis

Sample preparation was carried out using Ilumina Nextera^TM^ DNA Sample Prep Kits according to the manufacturer’s protocol. The amount of DNA was measured on a Qubit 2.0 fluorometer (Life Technologies, Carlsbad, CA, USA). Full exome sequencing was performed by selecting specific DNA fragments using the SureSelect system, followed by parallel sequencing of the resulting libraries applying Illumina technology. Sequencing of DNA fragments was carried out on the Ilumina Genome Analyzer HiSeq 2000 system. All sequences (reads) were aligned to the reference genome using the Burrows-Wheeler Alignment (BWA) program [10]. The human genome sequence (Genome Reference Consortium Human Build 37 (GRCh37-hg19)) was used as a reference. To address the bioinformatics challenges of exome data analysis we use the Best Practices workflow of GATK (Genome Analysis Toolkit from Broad Institute) [11].

Changes, detected by WES, were annotated in ANNOVAR program, using the summarize_annovar.pl script [12]. It makes possible to compare single nucleotide substitutions with a number of specialized databases and predict the functional significance of the detected changes using in silico tools (SIFT, PolyPhen-2, LRT, Mutation Taster, Mutation Assessor, ClinVar, phyloP, GERP++ and others) from dbNSFP v.1.3. In addition, the CADD (Combined Annotation Dependent Depletion) program was used. To search for somatic mutations in DNA samples, isolated from tumor tissue, the COSMIC database (catalog of somatic mutations in cancer) was employed. To estimate the population frequencies of the identified variants, we used data from the 1000 Genomes project, ESP6500, and the Exome Aggregation Consortium.

#### 2.2.2. Confirmation of Mutations

The Sanger sequencing method was used to confirm the molecular genetic changes identified as a result of WES. Preparation of matrices for sequencing included preliminary amplification of the desired DNA fragments using specific primers: F: 5′-CGGCTTGGGGAGATTGC-3′ and R: 5′- CGAGCTAGCACTTCTCGC -3′ for *VEGFA*, F: 5′- CTGCCCAGGGATAATCACTG-3′ and R: 5′- CAGTGAGCAGTAGAAGGAC-3′ for *FANCA*, F: 5′- GACAGTGAGATCTTATCTCAAAAGAAC-3′ and R: 5′- GTATTTTTAGCAGAGACTCAAACTCC-3′ for *CDH1*. The resulting PCR product was purified using the ExoSAP-IT^TM^ Express PCR Product Cleanup kit (Applied Biosystems). Determination of nucleotide sequences was carried out on an automatic sequencer Genetic Analyzer 3500 Applied Biosystems. The resulting chromatograms were read using SnapGene Viewer software (v. 6.2).

#### 2.2.3. Search for Described Variants in a Larger Cohort of Samples

Screening for mutations in the *FANCA* and *CDH1* genes was performed using melting curve analysis (HRM) using Eva Green dye. PCR cycling and HRM analysis was performed on the Rotor-Gene 6000^TM^ (Corbett Research, Mortlake, New South Wales, Australia). Primers were used: F: 5′-ATCCAGAGCAGATAAAATCCCCC-3′ and R: 5′- CAAGCGGCCCAGGAACTTAC-3′ for *FANCA*, F: 5′-CACCACAAATCCAGTGAACAACG -3′ and R: 5′-GGTATGAACAGCTGTGAGGATG-3′ for *CDH1*. Positive and negative controls were included in each experiment, and then all samples with a change in melting curves were further resequenced.

## 3. Results

WES of DNA samples, isolated from the tumor and non-tumor-bearing healthy gastric tissue of patients with gastric cancer (GC), displayed an average of 33,205 changes in the nucleotide sequence per sample in the tumor tissue, and 33,070 in normal tissue. At the same time, more than half of these changes were annotated as located in intron or 3′, 5′- non-translated regions.

Analysis of the identified genetic variants in DNA samples from healthy tissue revealed that majority are single-nucleotide substitutions with 49.35%—synonymous and 44.97%—non-synonymous variants, respectively. 0.54% of all uncovered mutations are lead to the formation or elimination of stop codons (0.51%—“stop-gain”, 0.03%—“stop-loss”); 1.13%—mutations lead to a shift in the reading frame; and 1.49%—insertions and deletions that do not lead to a shift of the reading frame. In DNA samples from tumor tissue, most of the detected variants were also non-synonymous (44.75%) and synonymous (47.67%) substitutions; 0.94%—mutations leading to the formation or elimination of stop codons (0.90%—“stop-gain”, 0.04%—“stop-loss”), 2.28%—mutations leading to a shift and 1.89%—insertions and deletions that do not lead to a shift in the reading frame (Figure 1).

The quantitative distribution of variants mapped in exons and splicing sites regions, according to their functional significance, is presented in Table 2 (Table 2).

After ANNOVAR annotation [12] the third stage of processing was conducted—search for pathogenic variants that may represent driver mutations in the development of gastric cancer. This further analysis included the use of a custom filters, based on the following criteria:

1. The selection of variants located in exons and splicing sites,

2. Selection of potentially functionally significant genetic variants: truncating variants (frame shift mutations and mutations leading to the formation of a stop codon) and non-synonymous single nucleotide substitutions,

3. Selection of variants with frequency no more than 1%, according to 1000 genomes, ESP6500 and Exome Aggregation Consortium. Previously undescribed variants with unknown frequency were not rejected at this stage if they had potential functional significance.

Using our somatic data cohorts, we show some of the key visualizations generated using Maftools R Bioconductor package (https://www.bioconductor.org/packages/devel/bioc/vignettes/maftools/inst/doc/maftools.html (accessed on 11 January 2022)) (Figure 2). We have generated the plots removing mutations with <0.01 frequency, we also removed mutations in introns, 5′ flank and 3′ flank regions and removed from the plot genes that are highly mutated like *MUC* genes (in which mutations of all types were observed). Figure 2a shows oncoplot, displaying the most mutated genes in all tumor samples, with genes sorted by mutational frequency. The transition and transversion plot (Figure 2b) summarizes SNVs into six categories.

Analysis of signaling pathways, containing genes selected according to the previous selection criteria was the next step of data processing. In the first line the signaling pathways involved in the occurrence/development of cancer in humans (DNA repair, apoptosis, cell cycle control, inflammation and immune response) and interacts partners of identified candidate gene, were considered. In addition, candidate genes associated with the risk of gastric cancer according to the GWAS studies were included, as were genes with pathogenic variants, identified in exome studies in GC patients in other countries. To collect this information, NCBI, COSMIC databases with information on genes (function, partners and interactions, participation in cell life processes, association with diseases) were employed.

Using specialized software, the impact of the detected genetic variants on the function of the protein was assessed and the role of the detected insertions/deletions was determined.

In the course of an exploratory analysis of the WES results, all identified genetic changes found in normal and tumor tissue, were evaluated for functional significance using specialized programs and public databases. For new variations, bioinformatic approaches have been applied.

The general sequence to determine the functional significance of unknown genetic changes included the following criteria: the position of a variable in the coding sequence, leading to an amino acid substitution or affecting the reading frame; in a regulatory region with potential significant impact on the function in the region of non-coding RNA, etc. Specialized databases, employed on this stage include:

dbSNP—database containing descriptions of single nucleotide polymorphisms, short insertions and deletions, short tandem repeats;

Reference Sequence (RefSeq)—this database allows to distinguish if substitution is in the coding or non-coding part of the genome with gene end exon annotation and possible functional impact: an amino acid change or a reading frame shift;

Sift—allows the user to predict the consequence of non-synonymous mutations for a protein sequence, encoded by a gene containing a replacement;

PolyPhen-2, LTR, MutationTaster annotation, MutationAssessor annotation, FATHMM annotation—databases that allow predicting the impact of a replacement on the structure and function of proteins;

CADD is a tool for assessing the deleteriousness of single nucleotide variants and insertions / deletions in the human genome;

ClinVar—a database containing information on clinically significant polymorphisms and their association with diseases;

1000 Genomes, esp6500, ExAC—databases containing information on the minor alleles frequencies, based on the results of genome and exome sequencing within the framework of the 1000 genomes and NHLBI Grand Opportunity Exome Sequencing Project.

Conducted analysis revealed three pathogenic mutations in GC patients: *CDH1* (c.1320+1G>A), *VEGFA* (c.27_28insCCCAGCCCCAGCTACCA, p.A9fs) and *FANCA* (c.G1874C, p.C625S), one of which has not been previously described. The characteristics of the selected options are presented in Table 3 (Table 3). All mutations were confirmed by Sanger sequencing (Figure 3, Figure 4 and Figure 5).

We conducted a further search for all three described variants in a larger cohort of samples. Search c.27_28insCCCAGCCCCAGCTACCA variant in the VEGFA gene was carried out using Sanger sequencing of the corresponding DNA region. The material for this study was DNA samples of tumor tissue of the stomach of 30 patients and DNA samples from peripheral blood of 90 patients with GC and 90 healthy individuals. Insertion of 17 nucleotides was not found in any of the studied samples (Figure 4b).

For c.1320+1G>A in the *CDH1* gene and c.G1874C in the *FANCA* gene we used DNA samples isolated from the peripheral blood of 200 patients with gastric cancer and 200 healthy donors, as well as 70 DNA samples from the tumor tissue of the stomach of patients with cancer. Mutation screening was performed using HRM-analysis followed by Sanger sequencing of samples with differences in melting temperature.

As a result of HRM-analysis of the *CDH1* gene locus, we identified only one DNA sample from the blood of a patient with gastric cancer with differences in melting temperature (Figure 6).

The patient was a 72-year-old woman with an established diagnosis of gastric cancer. Postoperative histological examination showed a highly differentiated adenocarcinoma of gastric cancer, which allows us to classify the patient’s intestinal type of gastric cancer. This sample was subsequently sequenced according to Sanger in order to verify the changes. As a result of sequencing, the genetic variant c.1320+1G>A of the CDH1 gene was not confirmed, but another change was detected nearby—this is the missense variant rs587781783 (c.1300G>C) in exon 9 of the CDH1 gene (Figure 7). This change was previously identified by a number of researchers in GC and registered in dbSNP (rs587781783, frequency in the European population 0.007%). This sequence change replaces glycine, which is neutral and nonpolar, with arginine, which is basic and polar, at codon 434 of the CDH1 protein (p.Gly434Arg.). Algorithms developed to predict the effect of missense changes on protein structure and function (SIFT, PolyPhen-2, CADD) suggest that this variant is likely to be pathogenic. But the available data are currently insufficient to determine the role of this variant in the disease. ClinVar classifies it as variant of uncertain significance.

HRM-analysis of the *FANCA* gene fragment revealed 6 samples with a change in melting temperature, 4 of them were from the group of patients and 2 from the control group. All these DNA samples were isolated from the peripheral blood of the examined individuals (Figure 8). All of these samples were confirmed by Sanger sequencing. In one of the GC patients the variant c.G1874C was detected as the homozygous (Figure 9).

Among patients with GC who had a change c.G1874C in the *FANCA* gene, there was an 88-year-old Russian man with an intestinal type GC, a 53-year-old Ukrainian man with a diffuse type of disease, a 76-year-old Russian woman with intestinal gastric cancer, and a 54-year-old man of Tatar ethnicity with a diffuse type of GC (he had the CC genotype of c.G1874C). Among healthy donors, this genetic variant was found in two men aged 55 and 61, both of Russian ethnicity. Thus, the frequency of allele C in patients was 1.25%, among healthy individuals 0.5%.

## 4. Discussion

WES, together with whole genome sequencing have become central methodologies for cost-effective detection of pathogenic genetic variations such as SNPs and insertions or deletions (Indels). Uncovering of novel genetic risk variants, using these technologies led to genotype-phenotype associations for many tumor types; including GC. However, there is still a need for further genetic studies in different populations, since only a part of gastric tumors could be explained by mutations in known or recently described genes. In our exploratory study on GC tumor and non-tumor-bearing healthy normal gastric tissue of patients with gastric adenocarcinoma WES approach was employed to identify new genes and genetic variants, associated with GC development. The workflow was divided in three main stages, which were performed sequentially: data pre-processing (reads have been aligned to a reference genome), variants discovery and functional annotation. There are some challenges in last two workflow steps, which are crucial for the interpretation of the data. The variant discovery step is essential in identifying the variation sites relative to the reference sequence, and important in calculating genotypes at that site for each sample. The challenge here is to balance the sensitivity and specificity. Sensitivity is critical to minimize false negatives, namely failing to identify real variants. Specificity, in turn, is important to minimize false positives, or failing to reject artifacts, since some of the observed variations are usually caused by mapping and sequencing artifacts. The next major challenge of WES approach is to verify the causative variant among a significant number of bystander signals that do not play any role in the disease etiology. Our strategy to restrict the number of candidates included very strict criteria: rejection of variants not shared between cases; and rejection of common variants, listed in dbSNP or 1000 Genomes, as these are expected to be not damaging, as harmful mutations must be rare and focus was given to the possible functional impact of the variation. Our study revealed a number of variants in different genes with different functional impact, known to be involved in GC pathogenesis, and those which were not yet linked to GC. For instance, it should be noted that in all patients, the largest number of genetic variants in both tumor and healthy tissue was found in the genes of the mucin family—*MUC3A* and *MUC16*. Mucins are high molecular weight glycoproteins synthesized by a wide range of epithelial tissues, including stomach. Mucins play an important role in the pathogenesis of malignant tumors of various localizations. *MUC16* is the largest transmembrane mucin that plays an important role in metastasis, protecting tumor cells from cytotoxic reactions that occur when exposed to natural killer cells [13]. *MUC3A* is the main glycoprotein component of mucus secreted by mucosal epithelial cells and performs a protective function. There is evidence that the *MUC3A* gene is involved in the pathogenesis of GC, being aberrantly expressed in gastric tumor cells, and there has also been found an association with the disease severity [14,15].

In our study, one patient (1 GC) harbors two pathogenic variants: the c.27_28insCCCAGCCCCAGCTACCA variant in the *VEGFA* and c.G1874C (rs139235751) in the FANCA genes. This patient is a male, diagnosed at the age of 59 years with undifferentiated adenogenic GC, with ingrowth of the muscular membrane, which also makes it possible to classify the diffuse type of GC. The c.27_28insCCCAGCCCCAGCTACCA variant in the *VEGFA* gene, not previously described in databases, was found in the heterozygous state only in the tumor tissue. This variant is located at the beginning of the first exon and leads to a frameshift (p.A9fs). The insertion of seventeen nucleotides is a duplication of the site c.28–44 of the first exon of the *VEGFA* gene (Figure 2). Since it was found in tumor tissue only and by the diffuse type of GC, it can be potentially considered as a marker of the severe course of the disease. The *VEGFA* (vascular endothelial factor A) gene encodes a member of the PDGF/VEGF family of growth factors. The protein functions as a glycosylated mitogen and acts on endothelial cells to increase vascular permeability, angiogenesis, vasculogenesis, endothelial cell growth and cell migration, and also inhibits apoptosis. Mutations in this gene are found in various types of cancer, including gastric cancer (https://www.mycancergenome.org/content/gene/vegfa/ (accessed on 1 January 2017). One of the methods of targeted therapy in gastric cancer is associated with the inhibition of angiogenesis, since cancer cells begin to stimulate angiogenesis in the early stages of oncogenesis, and angiogenesis stimulates tumor growth and progression [16]. VEGFA has been identified as the most potent cytokine involved in tumor angiogenesis and metastasis formation. Its activity is mediated by two tyrosine kinase receptors, VEGFR-1 and VEGFR-2, which differ significantly in their signaling properties. Some effects of VEGFA include increased vascular permeability [17], stimulation of serine protease or metalloprotease production [18,19], and inhibition of cell endothelial apoptosis [20]. Many studies have shown an increase in the expression of the *VEGFA* gene in the tissues of various solid tumors, including gastric cancer, a positive correlation has been established between aberrant expression of the *VEGFA* gene and the presence of metastases and a poor prognosis of the disease [21,22]. Raimondi A. and colleagues confirmed that *VEGFA* amplification can be used as a biomarker of long-term response to ramucirumab-based therapy in patients with metastatic gastric cancer, contributing to the personalization of treatment [23]. Our data of studying this variant on a large cohort of samples indicate that perhaps this genetic variant can be classified as variant of uncertain significance. However, studies are needed on large cohort of samples and in other populations of the world to understand whether this allele frequency is a feature of the populations of our region.

The second variant in the same patient c.G1874C (rs139235751) was in the *FANCA* gene, which is identified as pathogenic in all databases used, was found in the heterozygous state in both tumor and normal tissue that suggests its germinal status. The *FANCA* (Fanconi anemia, complementation group A) gene is a gene that codes for Fanconi anemia complementation group member protein. Fanconi anemia is a clinically and genetically heterogeneous recessive disorder which causes cytogenetic instability, hypersensitivity of DNA to crosslink agents, increased chromosome breakage, and impaired DNA repair. Various mutations in *FANCA* gene are also observed in cancers such as endometrial cancer, colon cancer, and gastric cancer (https://www.mycancergenome.org/content/gene/fanca/ (accessed on 1 January 2017)). The missense mutation of this gene found in our study leads to the substitution of p.C625S amino acids. According to Clinvar (https://clinvarminer.genetics.utah.edu/ (accessed on 1 January 2017)) in the ESP NHLBI exome sequencing project, C625S was observed in 27/8600 (0.31%) alleles in individuals of European ancestry, indicating that it may be a rare variant in this population. The C625S variant is a non-conservative amino acid substitution that can affect the secondary structure of a protein, as these residues differ in polarity, charge, size, and/or other properties. This substitution occurs at a position that is conserved across species, and in silico analysis predicts that this variant will likely damage protein structure/function.

The genetic variant c.1320+1G>A, identified in a second patient (8 GC), was found in the heterozygous state at position +1 of the donor splicing site of exon 9 of the *CDH1* e-cadherin gene. This is an identified gene and was described in detail earlier, being one of the main studied candidate genes for GC and associated with hereditary gastric cancer. The change c.1320+1G>A was found in the DNA isolated from the tumor tissue of this male patient (aged 67 at diagnosis) with poorly differentiated adenocarcinoma, which allows classifying the diffuse type of GC. In addition, the patient 8GC had metastases of carcinoma in the lymph nodes of the lesser omentum and tumor germination in the serous membrane and greater omentum, which are characteristic of a severe course of the disease. The change c.1320+1G>A is not registered in the dbSNP database, however, in the COSMIC database there is the information that this variant was detected and classified as a somatic one. This variant was described in minimum 6 patients (4 with diffuse type of gastric cancer, 1 with malignant appendix and 1 with breast cancer) as a result of various genome-wide studies: Genomic Mutation ID COSV55730311; Legacy Identifier COSM2996774 [24,25,26,27]. There is evidence that the c.1320+1G>A mutation affects the canonical donor motif AGgt at the junction of exon 9 and intron 9, which is critical for splicing [28]. Ghoumid J. and colleagues found a substitution of guanine for cytosine (c.1320+1G>C, rs886039685) in the described position of the *CDH1* gene in patients with blepharoylodontic syndrome [28,29]. The authors performed a functional analysis of the mutation and found that it leads to a deletion of the exon 9 transcript, which linked to the removal of most of the EC3 domain of the *CDH1* protein (61 amino acids from the Tyr380 residue to the Lys440 residue), presumably disrupting its adhesive function [28]. Kievit A. and colleagues found c.1320+1G>A and c.1320+1G>T variants in patients with blepharoylodontic syndrome. The authors believe that these mutations also lead to alternative splicing and deletion of exon 9 of the *CDH1* gene [29].

## 5. Conclusions

WES in our study revealed three mutations in two patients with diffuse gastric cancer. Search for the mutations in *CDH1*, *VEGFA* and *FANCA* genes can have an importance for GC. The somatic variant found in the *CDH1* gene also confirms the need to search for not only germline but also somatic mutations as potential markers of the severity of the disease.

## Figures and Tables

**Figure 1 genes-14-00280-f001:**
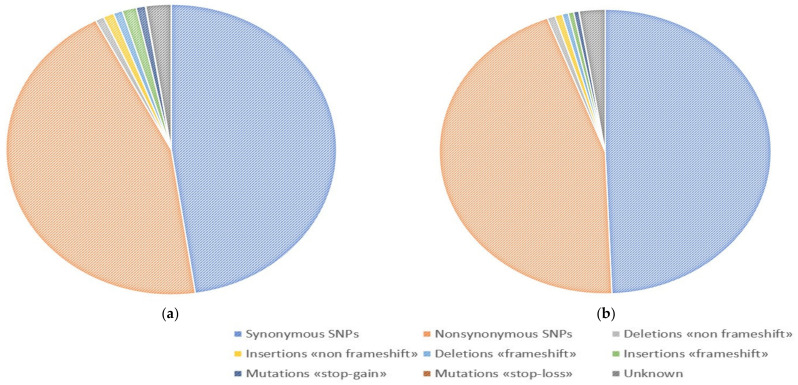
Distribution of variants mapped in exons and splicing sites regions, according to their functional significance (**a**) Tumor tissue; (**b**) Normal tissue.

**Figure 2 genes-14-00280-f002:**
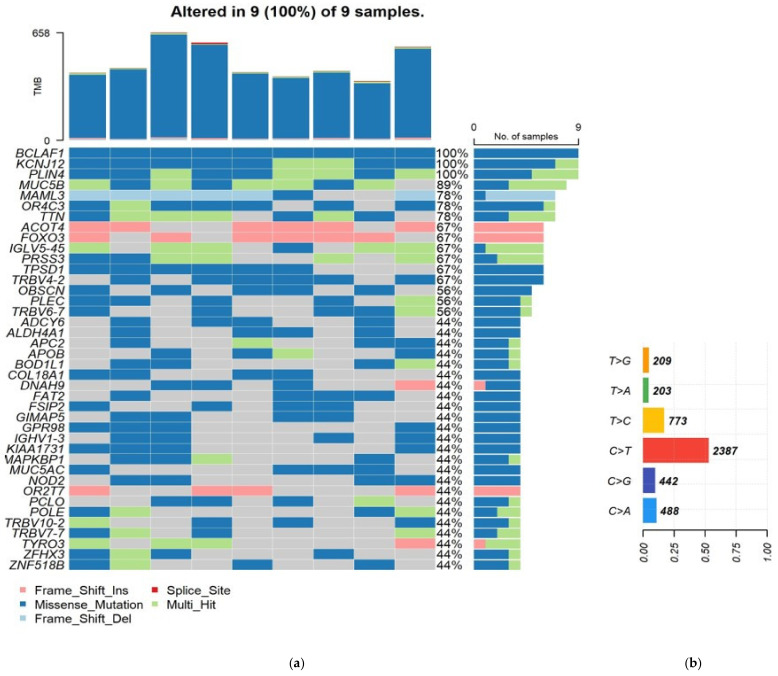
Key plots generated by Maftools visualization module (**a**) Oncoplot displaying the somatic landscape of GC cohort; (**b**) Transition and transversion plot displaying distribution of SNVs in GC classified into six transition and transversion events.

**Figure 3 genes-14-00280-f003:**
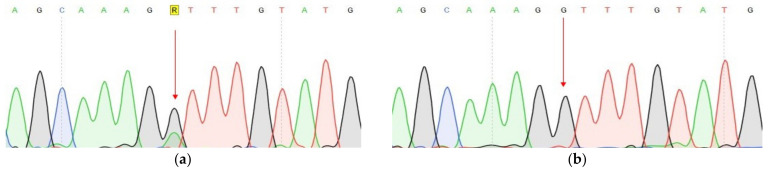
Fragments of the DNA sequence with the c.1320+1G>A variant in the *CDH1* (**a**) DNA sample with a change, isolated from the tumor tissue of a patient 8 GC; (**b**) Control DNA sample without change.

**Figure 4 genes-14-00280-f004:**
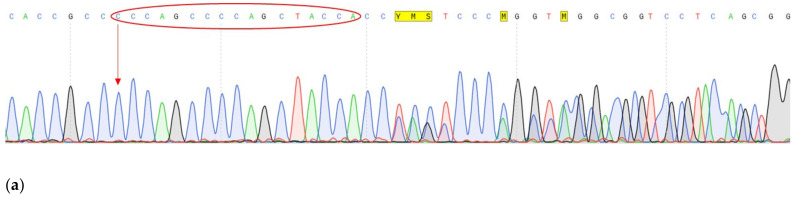
Fragments of the DNA sequence with the c.27_28insCCCAGCCCCAGCTACCA variant in the *VEGFA* gene. (**a**) DNA sample with a change isolated from the tumor tissue of a 1 GC patient; (**b**) Control DNA sample without change.

**Figure 5 genes-14-00280-f005:**
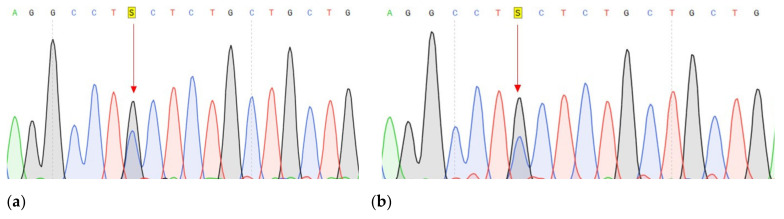
Fragments of the DNA sequence with the c.G1874C variant in the *FANCA* gene. (**a**) DNA sample with a change, isolated from the tumor tissue of a patient with 1 GC; (**b**) DNA sample with a change isolated from normal tissue of a 1GC patient; (**c**) Control DNA sample without change.

**Figure 6 genes-14-00280-f006:**
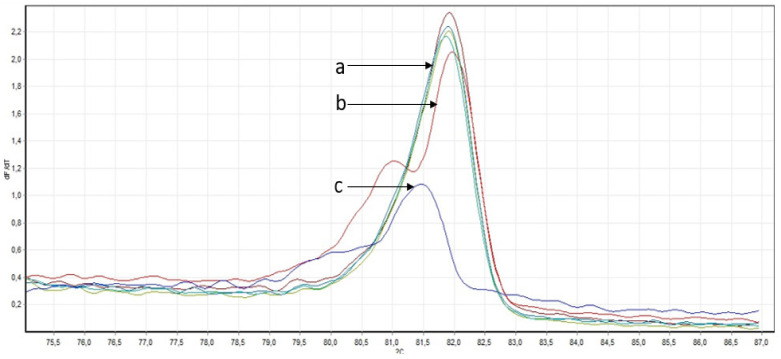
Derivatives of melting curves of amplicons obtained during PCR of a section of the *CDH1* gene. (**a**) DNA samples without changes in the nucleotide sequence; (**b**) DNA sample with difference in melting point; (**c**) DNA sample with c.1320+1G>A (positive control).

**Figure 7 genes-14-00280-f007:**
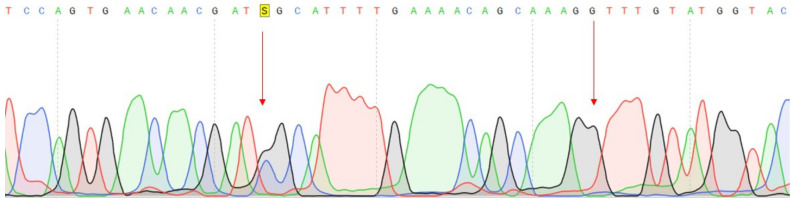
Fragments of the DNA sequence locus in the *CDH1* gene (DNA sample with difference in melting point).

**Figure 8 genes-14-00280-f008:**
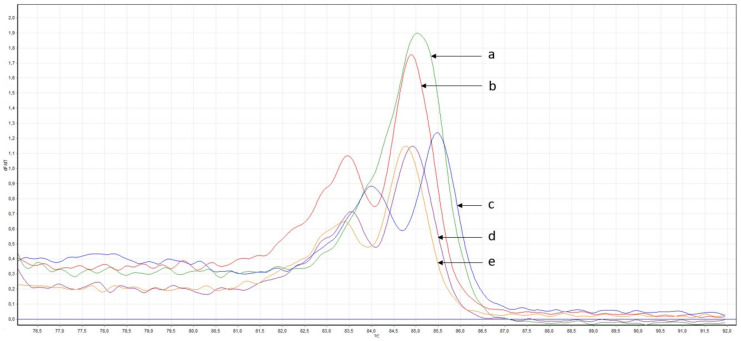
Derivatives of melting curves of amplicons obtained during PCR of a section of the *FANCA* gene. (**a**) DNA samples without changes in the nucleotide; (**b**,**d**,**e**) Example of DNA samples with difference in melting point; (**c**) DNA sample with c.G1874C (positive control).

**Figure 9 genes-14-00280-f009:**
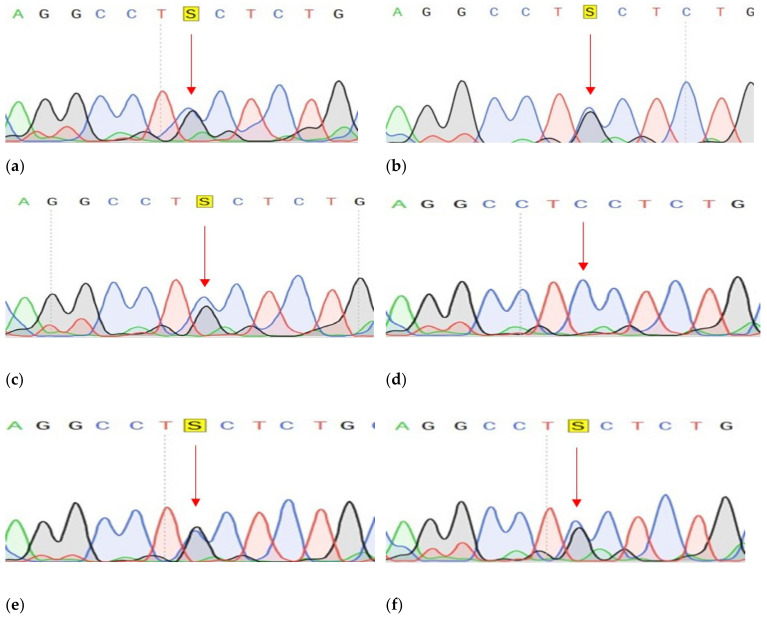
Fragments of the DNA sequence locus in the *FANCA* gene (DNA sample with difference in melting point): (**a**–**d**) Patients with GC; (**e**,**f**) healthy donors.

**Table 1 genes-14-00280-t001:** Clinical and pathological characteristics of patients with gastric cancer.

Sample	Age	Sex	Cancer Stage	Histological Examination of the Tumor
1 GC	59	m	3	Undifferentiated adenogenic gastric carcinoma with invasion of the muscularis
2 GC	69	f	3	Undifferentiated adenogenic gastric cancer with invasion of the muscular layer
3 GC	77	f	3	Poorly differentiated adenocarcinoma of a scirrhous structure, germinating all layers of the stomach wall with the presence of perineural invasion
4 GC	54	m	3	Surface ulcerated malignant tumor
5 GC	61	m	3	Moderately differentiated adenocarcinoma invading all layers of the stomach wall
6 GC	45	m	3	In the submucosal layer, a poorly differentiated adenocarcinoma of the stomach, growing into the muscle layer
7 GC	65	m	3	Undifferentiated adenogenic gastric cancer growing into subserosis
8 GC	67	m	3	Ulcerated on the surface, low-grade adenocarcinoma of the stomach with germination of the muscular membrane and the presence of cancer emboli in the vessels
9 GC	74	m	3	Moderately differentiated adenocarcinoma of the stomach, sprouting all layers of its wall with the presence of perinephric invasion

m—male; f—female.

**Table 2 genes-14-00280-t002:** Quantitative distribution of variants, located in exons and splicing regions, according to their functional significance.

**Sample**	**1 GC**	**2 GC**	**3 GC**	**4 GC**	**5 GC**	**6 GC**	**7 GC**	**8 GC**	**9 GC**
**Tumor tissue**									
Total number of variants	19,179	19,154	14,829	11,850	11,527	12,213	14,975	20,788	10,275
Synonymous SNPs	9584	9530	7377	5563	5755	5694	7519	8293	4939
Nonsynonymous SNPs	8623	8670	6650	5493	5103	5547	6662	9000	4573
Deletions «non frameshift»	136	127	116	106	111	99	119	237	93
Insertions «non frameshift»	118	114	105	98	88	137	107	552	84
Deletions «frameshift»	88	92	89	87	60	155	65	489	79
Insertions «frameshift»	68	65	48	84	65	163	63	1217	95
Mutations «stop-gain»	68	80	55	121	79	112	58	550	92
Mutations «stop-loss»	7	9	5	3	6	6	6	15	2
Unknown	487	467	384	295	260	300	376	435	318
**Sample**	**1 GC**	**2 GC**	**3 GC**	**4 GC**	**5 GC**	**6 GC**	**7 GC**	**8 GC**	**9 GC**
**Normal tissue**									
Total number of variants	20,200	21,068	12,140	12,576	15,478	14,848	9169	10,323	12,031
Synonymous SNPs	10,115	10,517	5967	6145	7758	7280	4349	5024	5930
Nonsynonymous SNPs	9087	9508	5443	5714	6929	6722	4135	4640	5309
Deletions «non frameshift»	133	143	108	99	117	89	94	99	99
Insertions «non frameshift»	124	140	91	92	99	96	92	92	99
Deletions «frameshift»	91	86	68	68	94	110	69	70	106
Insertions «frameshift»	93	95	59	59	65	65	94	52	105
Mutations «stop-gain»	71	84	71	75	54	84	81	53	77
Mutations «stop-loss»	7	9	2	5	4	5	2	2	3
Unknown	479	486	331	319	358	397	253	291	303

**Table 3 genes-14-00280-t003:** Characterization of selected pathogenic mutations.

Gen	*CDH1*	*VEGFA*	*FANCA*
**Tissue**	tumor	tumor	tumor, normal
**Chromosome**	Chr16	Chr6	Chr16
**DNA change**	c.1320+1G>A	c.27_28insCCCAGCCCCAGCTACCA	c.G1874C
**Protein change**	-	p.A9fs	p.C625S
**Frequency by 1000 Genomes**	-	-	0.001
**dbSNP**	-	-	rs139235751
**PolyPhen-2 ^1^**	-	-	D (0.986)
**LTR ^2^**	-	-	D
**Mutation Taster ^3^**	D	-	D
**Mutation Assessor ^4^**	-	-	M
**FATHMM ^5^**			D
**CADD ^6^**	34	23.5	26.4
**Sift ^7^**	-	-	D (0.006)

^1^ PolyPhen2: D-Probably damaging (>=0.957); ^2^ LTR: D: Deleterious; ^3^ MutationTaster: D-disease causing; ^4^ MutationAssessor: M-medium; ^5^ FATHMM—D: Deleterious; ^6^ CADD: the higher the value, the more pathogenic the variant, >20-deleterious; ^7^ Sift: D: Deleterious (sift <= 0.05).

## Data Availability

Raw data of WES available at the link https://www.ncbi.nlm.nih.gov/sra/PRJNA857496 (accessed on 11 July 2022).

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
