# Peer review of "Whole Exome Sequencing Study Suggests an Impact of FANCA, CDH1 and VEGFA Genes on Diffuse Gastric Cancer Development"

_genes, 2023, doi:10.3390/genes14020280_

Round 1

Reviewer 1 Report

Summary

In this manuscript, the authors have described statistical analysis of whole exome data from gastrointestinal cancer patients, in search of novel protein coding somatic mutations. The authors report three significant mutations occurring in the cohort of Russian cancer patients in the FANCA, CDH1 and VEGFA genes. The methodology used for the presented analysis is sound. There are a few small additions that can be made to improve the presentation of the results, and make it easier to interpret for the reader.

Recommended revision

  1. Some additional exploratory analyses on the identified mutations may reveal patterns relevant to the mutations identified in this analysis. The R/bioconductor package maftools offers several features to enable easy exploratory analyses, such as:
    1. Oncoplot to show the ocurrences of the specific mutations relative to each other in each patient.
    2. Somatic interaction plot to explore the co-occurence of the mutations in the 3 significant genes identified with mutations in other genes.
  2. Table 2 shows the mutation counts, their types and locations for the patients. This is somewhat cumbersome for the reader to interpret in the table form. A much better way to show the distributions would be in the form of pairwise barplots, for the tumor and normal tissue, for the various regions.

Author Response

Dear reviewer, let me thank you for the detailed analysis of our manuscript and valuable comments.

We tried to take into account your wishes and answer your questions. We also want to note that we made changes to English with the help of a native speaker colleague

Point 1: Some additional exploratory analyses on the identified mutations may reveal patterns relevant to the mutations identified in this analysis. The R/bioconductor package maftools offers several features to enable easy exploratory analyses, such as: 1. Oncoplot to show the ocurrences of the specific mutations relative to each other in each patient. 2. Somatic interaction plot to explore the co-occurence of the mutations in the 3 significant genes identified with mutations in other genes.

Response 1: Unfortunately, neither I nor my colleagues have ever worked with R/bioconductor package maftools before. It took us some time to study. We tried to use this tool with our data and integrate it into the manuscript (Figure 2 new). We have generated the plots removing mutations with <0.01 frequency, we also re-moved mutations in introns, 5'flank and 3'flank regions and removed from the plot genes that are highly mutated like MUC genes (in which mutations of all types were observed). Figure 2a shows oncoplot, displays most mutated genes in all tumor samples, genes are sorted by mutational frequency. The transition and transversion plot (Fig. 2b) summarizes SNVs into six categories.

Point 2: Table 2 shows the mutation counts, their types and locations for the patients. This is somewhat cumbersome for the reader to interpret in the table form. A much better way to show the distributions would be in the form of pairwise barplots, for the tumor and normal tissue, for the various regions.

Response 2:  We added pie charts that shows distribution in tumor and normal tissue of variants mapped in exons and splicing sites regions, according to their functional significance (Figure 1 new). However, we decided to leave table 2 as well, since it contains numbers on the occurrence of genetic variants, perhaps they will be of interest to readers

Reviewer 2 Report

The authors did the whole exome sequencing in this manuscript on nine GC and nine adjacent normal gastric tissue samples. The authors made a nice start to the study. But it is not enough for an original paper. 

Here are some major concerns:

1. The findings are limited to one or two gastric cancer patient samples. And there is no validation on other samples. 

2. There is no function study or mechanism study on the results. 

Since the sample number is limited and lacks validation/functional study, more data is needed to support the authors' findings. 

Author Response

Dear reviewer, let me thank you for the detailed analysis of our manuscript and valuable comments.

We tried to take into account your wishes and answer your questions. We also want to note that we made changes to English with the help of a native speaker colleague

Point 1: The findings are limited to one or two gastric cancer patient samples. And there is no validation on other samples. 

Response 1: In our study, we performed whole exome sequencing in 9 patients in two tissue samples (tumor and normal, total 18 samples). Variants (n=3) that were selected as pathogenic (based on their analysis in various databases described in the article) and also confirmed by Sanger sequencing were detected in only 2 patients, so we have discussed them in this article. We validated our results using Sanger sequencing in a total of 90 DNA samples (in which exome sequencing has not been performed): in other 30 patients with diffuse gastric cancer in both DNA samples isolated from blood leukocytes and isolated from tumor tissue (total 60 samples of DNA), as well as in 30 healthy donors. None of the variants c.1320+1G>A in the CDH1 gene,  c.27_28insCCCAGCCCCAGCTACCA (p.Ala9fs) of the VEGFA gene and c.G1874C (p.Cys625Ser) in the FANCA was identified in these samples.

Point 2: There is no function study or mechanism study on the results. Since the sample number is limited and lacks validation/functional study, more data is needed to support the authors' findings.

Response 2:  

As we wrote in an answer to a previous question, we screened the detected genetic variants in DNA samples from 30 gastric cancer patients and 30 healthy donors. In the future, we plan to search for these options on a larger sample (more than 500 people) and plan to present these results later. Unfortunately, we do not currently have the opportunity to conduct functional studies of the detected mutations; nevertheless, in selecting them, we relied on data from a larger number of predictive databases. We consider the results of this study to be important for further study and identification of important genes that drive the development of gastric cancer, since the CDH1 gene is very well known for gastric cancer, while the FANCA and VEGFA genes have not been studied enough in gastric cancer and can be included in a targeted panel of oncomarkers specific for this type of cancer.

Also we added pie charts that shows distribution in tumor and normal tissue of variants mapped in exons and splicing sites regions, according to their functional significance (Figure 1 new). Using our somatic data cohorts, we show some of the key visualizations generated using Maftools R Bioconductor package (Figure 2 new).
